# Polychaete Diversity Related to Different Mesophotic Bioconstructions along the Southeastern Italian Coast

**Maria Flavia Gravina** [1,2,*], **Cataldo Pierri** [2,3], **Maria Mercurio** [2,3], **Carlotta Nonnis Marzano** [2,3] **and Adriana Giangrande** [2,4]

1   Department of Biology, University of Rome "Tor Vergata", 00133 Rome, Italy
2   CoNISMa, Consorzio Interuniversitario per le Scienze del Mare, 00196 Rome, Italy
3   Department of Biology, University of Bari Aldo Moro, 70125 Bari, Italy
4   Department of Biological and Environmental Sciences and Technologies, University of Salento, 73100 Lecce, Italy
*   Correspondence: maria.flavia.gravina@uniroma2.it

**Abstract:** In the different mesophotic bioconstructions recently found along the Southeastern Italian coast, polychaetes have been proved to show high species richness and diversity, hitherto never investigated. In the present study, the species composition and functional role of polychaete assemblages were analysed; the updated key to identification of the Mediterranean species of genus *Eunice* was presented and some taxonomic issues were also discussed. On the total of 70 species Serpulidae and Eunicida were the dominant polychaetes. Facing similar levels of α-diversity, the polychaete assemblages showed a high turnover of species along the north-south gradient, clearly according to the current circulation pattern, as well as to the different bioconstructors as biological determinants. Indeed, Serpulidae were dominant on the mesophotic bioconstructions primarily formed by the deep-sea oyster *Neopycnodonte cochlear*, while the Eunicida prevailed on the mesophotic bioconstructions mainly built by scleractinians. Lastly, the record of *Eunice dubitata* was the first for the Mediterranean and Italian fauna and proved this species to be characteristic of mesophotic bioconstructions.

**Keywords:** Polychaete Eunicida; Polychaete Serpulidae; marine bioconstructions; polychaete diversity; mesophotic bioconstructions; Mediterranean Sea; Southeastern Italian coast; Italian fauna

## 1. Introduction

Mediterranean polychaetes are proved to be good bioindicators of environmental conditions and ecological status both on sedimentary and rocky bottoms [1–3]. Such results are achieved following investigations especially in shallow habitats, as well as in circalittoral habitats where coralligenous formations occur [4–6].

The coralligenous is a characteristic Mediterranean biocoenosis which is an object of detailed studies, due to its role in shaping the seascape, formed by perennial algae and animal organisms with consistent calcareous concretions in sciaphilic environments, from 20 to 120 m depth [7–11]. Moreover, particular attention has recently been focused on more deep-sea habitats, such as sea mountains, non-symbiotic coral reefs, and submarine canyons, where peculiar invertebrates, mainly scleractinians, gorgonians, and antipatharians, act as the main habitat formers [12–18]. Also, the habitats located in the twilight zone, so-called "mesophotic zone" ranging from 30–40 to 150 m depth, are currently under study. Unfortunately, such habitats are still poorly investigated, and some possible confusion exists in the definition of the mesophotic zone (see Cerrano et al. [19]). Few recent studies have been conducted in the Ligurian Sea and Tyrrhenian Sea [20,21], as well as in the Southern Adriatic off the Italian coast [22–24].

The peculiar mesophotic communities recently found along the Apulian coast were described by Corriero et al. [22] and Cardone et al. [24], paying particular attention to

their characterization and underlining their crucial ecological role in supporting high habitat complexity and enhancing local biodiversity. Only secondary to the primary bioconstructors of such bioconstructions, polychaetes have been proved to be a very rich and diverse component of such mesophotic communities. This group is very interesting because includes both vagile and sessile species, which may play diverse roles in forming the bioconstruction architecture and contribute to its functioning. Among the vagile forms the components of the family Eunicidae are known to be particularly relevant, some species of which are considered symbiotic with corals [25,26].

In the present study the polychaete assemblages associated with the mesophotic bioconstructions recently discovered along the Apulian coast are considered, with special focus on the following objectives: (i) to analyse their diversity patterns in the different bioconstructions both in terms of species composition and functional role; (ii) to analyse the taxonomic issues within the family Eunicidae in order to clarify the statement of the species referred to the genus *Eunice*; (iii) to update the checklist of the Mediterranean and Italian fauna of the polychaetes referred to the genus *Eunice*.

## 2. Material and Methods

### 2.1. Study Areas

The three study areas are located along the Southeastern Italian coast (Adriatic and Ionian coast of Apulia) (Figure 1), and harbour three different bioconstructions recently discovered and described [22,24].

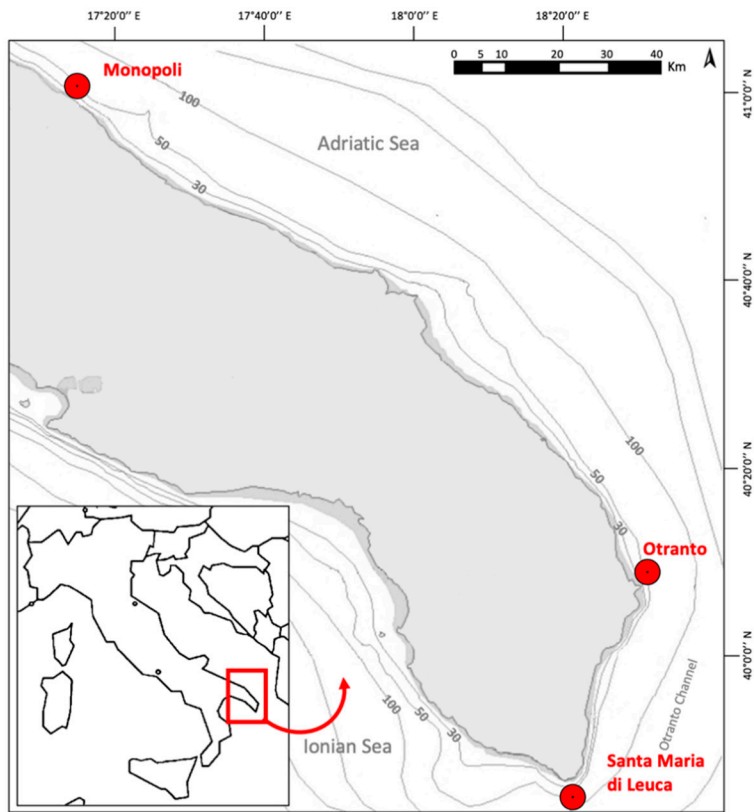

**Figure 1.** Location of the three areas under study off the Southeastern Italian coast (Apulia).

Monopoli site (MON) is the northernmost and located approximately 1.5 nautical miles off the coast in the Southern Adriatic Sea. At this site the bioconstruction is up to 2 m in thickness and occurs in a depth range between 30 and 55 m, along a fault line with NW-SE orientation. The bioconstruction is characterized by two species of non-symbiotic scleractinians *Phyllangia americana mouchezii* (Lacaze–Duthiers, 1897) and *Polycyathus muellerae* (Abel, 1959) as primary bioconstructors. The site of Otranto (OTR)

is located at about 1 nautical mile off the coast in the Otranto Channel, which marks a passage from the Adriatic to the Ionian Sea, within the bathymetric range 45–64 m; the site of Santa Maria di Leuca (SML) is placed approximately at 1.5 nautical miles off the coast in the Ionian Sea within a 45–70 m depth range. In both these latter sites the bioconstructions are mainly built by the bivalve *Neopycnodonte cochlear* (Poli, 1795) and give rise to different structures like pinnacles and globose formations. The structural complexity of the bioconstruction at all the sites strongly supports a high level of biodiversity [22,24].

### 2.2. Sampling Methods and Taxonomic Analysis

Samplings were performed during the period August–September, in 2017 at MON and in 2018 at OTR and SML. The seafloor was dominated by a fault perpendicular to the coastline up to 55 m depth at MON and by a steep slope connecting the upper zone to the deeper areas, 64 m and 70 m, respectively, at OTR and SML. The coastal dynamics were characterized by a wave-dominated microtidal setting and the wave conditions produced prevailingly coastal longshore currents with NW–SE direction. For the characterization of the benthic assemblages, three samples (approximately three liters of total volume) were scraped from the bioconstruction at each study area by technical divers at 45–50 m depth, according to the sampling methods described by Corriero et al. [22] and Cardone et al. [24]. The collected biological material was sorted in the laboratory, and all the specimens were preserved in an alcohol solution and then identified at the lowest possible taxonomical level, using Zeiss stereo and optical microscopes. In the present study, the polychaete component of the benthos has been considered.

For the polychaete's faunal analysis, the vagile specimens were extracted from the samples and their position within the bioconstruction was observed and recorded. Similarly, the aggregation distribution, as well as the growth mode and superimposition of the calcareous tubes of the sessile worms were noticed and reported, with the aim to understand the role played by the polychaetes with respect to the bioconstruction organization and functioning. The polychaete specimens preserved in the authors' collections and in the collection of the Museum of Porto Cesareo Zoological Laboratory (PCZL), University of Salento, Italy, were also analysed and compared with the material of the present samples, in order to disentangle some criticisms about the identification of the Eunicidae. The following complete revisions and past and current significant literature regarding this family were also examined [27–38], with the purpose of producing a revised and updated dichotomous key for the identification of the Mediterranean species of the genus *Eunice sensu lato*, within the family Eunicidae.

### 2.3. Data Analysis

Polychaete diversity was measured in terms of species richness (α-diversity) and species turnover along the local geographical North-South gradient (β-diversity). This latter was computed using the Whittaker Index $\beta_w = (S/\bar{a}) - 1$, where S is the total number of species that results from merging the number of species of each pairwise of sites considered and $\bar{a}$ is the average number of species per each pairwise of a sample [39,40]. The species similarity between polychaete assemblages of the investigated sites was measured by means of the Sörensen Index.

## 3. Results

### 3.1. Species Composition and Diversity

A total of 70 species were found, most of them belonging to the family Serpulidae (54.3% with 38 species) and to the order Eunicida (23% with 15 species), this latter being mostly represented by species of the family Eunicidae accounting for 10 species (Table 1). Among them, *Eunice dubitata* constitutes a new record for the Italian fauna and amends the checklist [41].

**Table 1.** List of polychaete species recorded at three sites along the Apulian coasts.

| Site | | Monopoli | Otranto | Santa Maria di Leuca |
|---|---|---|---|---|
| **Depth**<br>**Main Structuring species** | | **43 m**<br>Scleractinians | **52 m**<br>*N. cochlear* | **60 m**<br>*N. cochlear* |
| **Family** | **Species** | | | |
| Sabellidae Latreille, 1825 | *Hypsicomus stichophthalmos* (Grube, 1863) | x | | |
| Serpulidae Rafinesque, 1815 | *Hydroides pseudouncinata* (Zibrowius, 1968) | x | x | |
| | *Serpula vermicularis* Linnaeus, 1767 | x | x | x |
| | *Serpula cavernicola* Fassari & Mollica, 1991 | x | x | x |
| | *Serpula concharum* Langerhans, 1880 | x | x | x |
| | *Serpula lobiancoi* Rioja, 1917 | | x | x |
| | *Serpula israelitica* Amoureux, 1977 | | | x |
| | *Spiraserpula massiliensis* (Zibrowius, 1968) | x | x | x |
| | *Vermiliopsis infundibulum* (Philippi, 1844) | x | x | x |
| | *Vermiliopsis striaticeps* (Grube, 1862) | x | x | x |
| | *Vermiliopsis monodiscus* Zibrowius, 1968 | x | | x |
| | *Vermiliopsis labiata* (O. G. Costa, 1861) | x | x | x |
| | *Bathyvermilia eliasoni* (Zibrowius, 1970) | | | x |
| | *Metavermilia multicristata* (Philippi, 1844) | x | x | x |
| | *Semivermilia agglutinata* (Marenzeller, 1893) | | x | x |
| | *Semivermilia crenata* (O. G. Costa, 1861) | x | x | x |
| | *Semivermilia cribrata* (O. G. Costa, 1861) | | x | x |
| | *Semivermilia pomatostegoides* (Zibrowius, 1969) | | x | x |
| | *Filogranula gracilis* Langerhans, 1884 | x | | x |
| | *Filogranula calyculata* (O. G. Costa, 1861) | | x | x |
| | *Filogranula annulata* (O. G. Costa, 1861) | | x | x |
| | *Janita fimbriata* (Delle Chiaje, 1822) | x | | x |
| | *Spirobranchus lima* (Grube, 1862) | | | x |
| | *Spirobranchus polytrema* (Philippi, 1844) | | | x |
| | *Spirobranchus triqueter* (Linnaeus, 1758) | x | x | x |
| | *Placostegus tridentatus* (Fabricius, 1779) | | x | x |
| | *Verminia cristallina* (Philippi, 1844) | x | x | x |
| | *Josephella marenzelleri* Caullery & Mesnil, 1896 | | x | x |
| | *Filograna implexa* Berkeley, 1835 | x | x | x |
| | *Protula tubularia* (Montagu, 1803) | | x | x |
| | *Spirorbis cuneatus* Gee, 1964 | | | x |
| | *Spirorbis marioni* Caullery & Mesnil, 1897 | | x | x |
| | *Protolaeospira striata* (Quiévreux, 1963) | | x | x |
| | *Janua heterostropha* (Montagu, 1803) | | x | x |
| | *Neodexiospira pseudocorrugata* (Bush, 1905) | | x | x |
| | *Pileolaria militaris* Claparède, 1870 | | x | x |
| | *Pileolaria heteropoma* (Zibrowius, 1968) | | | x |
| | *Vinearia koehleri* (Caullery & Mesnil, 1897) | | | x |
| | *Nidificaria clavus* (Harris, 1968) | | x | |
| Euphrosinidae Williams, 1852 | *Euphrosine foliosa* Audouin & H Milne Edwards, 1833 | x | x | x |
| Eunicidae Berthold, 1827 | *Eunice dubitata* Fauchald, 1974 | x | x | |
| | *Eunice schizobranchia* Claparède, 1870 | x | | |
| | *Eunice pennata* (Müller, 1776) | | | x |
| | *Eunice floridana* (Pourtalès, 1867) | | | x |
| | *Eunice torquata* (Quatrefages, 1866) | x | x | |
| | *Palola siciliensis* (Grube, 1840) | x | x | |
| | *Palola valida* (Gravier, 1900) | | x | |
| | *Paucibranchia fallax* (Marion & Bobretzky, 1875) | x | x | |
| | *Lysidice collaris* Grube, 1870 | x | x | x |
| | *Lysidice ninetta* Audouin & H Milne Edwards, 1833 | x | x | |
| Lumbrineridae Schmarda, 1861 | *Lumbrineris coccinea* (Renier, 1804) | x | x | |
| | *Scoletoma laurentiana* (Grube, 1863) | x | | |
| Oenonidae Kinberg, 1865 | *Arabella geniculata* (Claparède, 1868) | x | x | |
| | *Arabella iricolor* (Montagu, 1804) | x | | |
| | *Drilonereis filum* (Claparède, 1868) | x | x | |
| Glyceridae Grube, 1850 | *Glycera tesselata* Grube, 1863 | x | | |
| | *Glycera unicornis* Lamarck, 1818 | x | | |
| Goniadidae Kinberg, 1866 | *Glycinde nordmanni* (Malmgren, 1866) | x | | |
| | *Goniada emerita* Audouin & H Milne Edwards, 1833 | x | | |
| | *Goniada maculata* Örsted, 1843 | x | x | |
| Nereididae Blainville, 1818 | *Ceratonereis costae* (Grube, 1840) | x | x | |
| Polynoidae Kinberg, 1856 | *Harmothoe antilopes* McIntosh, 1876 | x | | |
| | *Harmothoe pagenstecheri* Michaelsen, 1896 | | x | x |
| | *Lepidasthenia elegans* (Grube, 1840) | x | | |
| Syllidae Grube, 1850 | *Haplosyllis spongicola* (Grube, 1855) | | x | |
| | *Syllis alternata* Moore, 1908 | x | | |
| | *Syllis gracilis* Grube, 1840 | x | x | |
| | *Syllis ferrani* Alós & San Martín, 1987 | | x | |
| | *Syllis variegata* Grube, 1860 | x | | x |
| | *Sphaerosyllis hystrix* Claparède, 1863 | | x | |

Species richness was highest at OTR, where 46 species were identified; whilst at SML and MON 42 and 40 were respectively found. However, when compared, sites showed a different species composition. Serpulids were the most responsible for the SML species richness, whilst eunicids for the richness of MON and OTR.

Among serpulids, at MON *Hydroides pseudouncinata*, a shallow and sciaphilic and coralligenous species, and *Serpula massiliensis*, a species characteristic of caves, were the most common species. Here the bioconstruction, mostly built by corals, was made in its basal layer by a dense intertwining of dead corallites and tubes of the largest serpulids, particularly *Serpula cavernicola* and *S. massiliensis*, which are typical inhabitants of submerged caves. Moreover, the exposed surface of the reef was extensively colonized by a high number of species: together with *S. cavernicola*, two main gregarious species, *S. massiliensis* and *Filograna implexa*, contributed significantly to the formation of the reef especially colonizing outlines and overhangs of the concretion: the first species was found with many intertwined tubes and the second one with assemblages which, even made by fragile and friable tubes, showed very large extension. Many serpulids were observed with their tubes in epibiosis on calcareous skeletons and valves of other sessile organisms, such as *Hydroides pseudouncinata*, *Spirobranchus triqueter*, *Serpula vermicularis*, *Serpula concharum* which are largely occurring also on shallow-shelf habitats; *Vermiliopsis monodiscus*, *Janita fimbriata*, *Metavermilia multicristata* which are deep-water and bathyal species; *Semivermilia crenata*, *Placostegus crystallinus*, *Filogranula gracilis*, *Vermiliopsis labiate*, which are characteristic of coralligenous and cave habitats.

At OTR and SML few highly dominant species of serpulids, including *Vermiliopsis infundibulum*, *S. crenata*, *Filogranula annulata* and *Semivermilia pomatostegoides*, occurred, and they were found together with some particularly abundant small species of spirorbids, such as *Protolaeospira (Protolaeospira) striata* and *Pileolaria militaris*. Different groups of species occurred in different microhabitats of the bioconstruction: species of shallow shelf (*Spirobranchus polytrema*, *Janua pagenstecheri*, *Pomatoceros triqueter*) and detritic bottoms (*Semivermilia cribrata* and *Spirorbis (Spirorbis) cuneatus*) mostly occurred on the outer edges of the reef, whilst species characteristic of deep cryptic and cave habitats (*Vermiliopsis monodiscus*, *Serpula israelitica*, *F. gracilis S. cavernicola*, *F. annulata*) were principally found within the *Neopycnodonte* valves and the reef interstices. Spirorbids showed particular adaptation to cryptic and deep crevices of the bioconstruction, as a result of their small dimensions and often-wrapped tubes; *P. (P.) striata*, *P. militaris* and *Vinearia koehleri* were also observed on the bare surfaces, such as the external edge and the smooth inner parts of the *Neopycnodonte* valves.

Among the vagile fauna, eunicids were the most abundant polychaetes at all the investigated sites: particularly *Eunice dubitata* and *Eunice torquata* were abundant especially at MON and OTR, whilst *Eunice pennata* and *Eunice floridana* were abundant at SML. In addition, *Lysidice collaris* was found abundant at all the sites, and the other abundant species *Palola siciliensis* and *Paucibranchia fallax* only occurred at MON and OTR.

Most of the specimens of *E. dubitata*, *E. torquata*, *E. floridana* and *P. siciliensis* were extracted from sinuous and twisted galleries entirely surrounded by corallites of scleractinians and *Neopycnodonte* valves; most of the complete specimens were large in dimensions, *E. dubitata* reaching a total length of 220 mm with 200 segments, *E. torquata* 180 mm with 210 segments, *E. floridana* 90 mm with 125 segments, respectively. Among the species of the genus *Eunice*, *E. torquata* and *E. pennata* are known from sciaphilic detritic and coralligenous habitats, while *E. dubitata* and *E. floridana* are reported from the deep to bathyal zone often associated to corals; contrarily, *P. siciliensis* and *P. fallax* were widely distributed both on infra-circalittoral detritic and shelf environments [5,42].

Lumbrineridae were also abundant even if represented by only two species. We paid particular attention to the identification of *Lumbrineris coccinea* which was found abundant at MON and OTR. This species, in fact, is often misidentified along the European coasts [32]. Our specimens were characterized by maxillary apparatus bearing five pairs of maxillae,

MI as long as MII, MIII clearly bidentate and by composite multidentate hooded hooks with short blade (Figure S1).

The polychaete α-diversity was similar at the three examined sites and ranged from 46 (OTR) to 40 species (MON). Conversely, the β-diversity varied from the highest values, 0.41 and 0.55, computed between MON and OTR and MON and SML respectively, to the lowest value, 0.33, found between OTR and SML. These results are in agreement with the measures of similarity which varied from a minimum of 0.43 between MON and SML and a maximum of 0.64 between OTR and SML (Table 2).

**Table 2.** Number of species (α), Sörensen Index (SI) and Whittaker Index (β) for the polychaete assemblages of the mesophotic bioconstructions along the Apulian coast.

| Sites | α | SI | β |
|---|---|---|---|
| MON (α)/MON-OTR (SI, β) | 40 | 0.59 | 0.41 |
| OTR (α)/MON-SML (SI, β) | 46 | 0.43 | 0.55 |
| SML (α)/OTR-SML (SI, β) | 42 | 0.64 | 0.33 |

*3.2. Taxonomic Accounts*

Polychaete members belonging to the genus *Eunice sensu lato* were particularly abundant in the mesophotic bioconstructions of the investigated Southeastern Italian coast, Apulian, and the collected material was also used for choosing the best diagnostic characters for the identification of the Mediterranean deep-water species. In this study the species found associated with the Mediterranean mesophotic bioconstructions were *E. torquata, E. dubitata, E. floridana, E. pennata*, while *E. norvegica* was found associated with deep white corals [16,17]. Shape and articulation of the antennae was one of the best characters to separate such species (Figure 2).

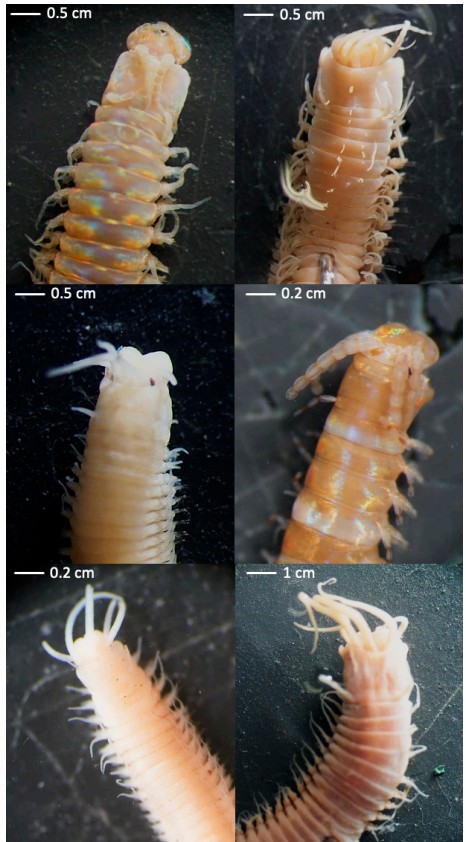

**Figure 2.** Anterior end in dorsal view of the species of the genus *Eunice sensu lato* found in the mesophotic bioconstructions investigated in the Southeastern Italian coast: upper line *E. dubitata* (left), *E. floridana* (right); middle line *E. schizobranchia* (left), *E. torquata* (right); bottom line *E. pennata* (left) *E. norvegica* (right).

The specimens of the five eunicid species collected in the present work were compared with the material of the Mediterranean species preserved in the PCZL collection (Porto Cesareo Zoological Laboratory, University of Salento, Italy) and in the personal collections of two authors. Based on the comparison and on the dated and current literature on the subject, a new updated key to identification of *Eunice* species of the Mediterranean is here proposed. The following diagnostic characters were considered for the key: shape and length of the antennae, peristomial and dorsal cirri, shape and disposition of the branchiae, chaetal morphology and arrangement, colour pattern. For the identification procedure all the species have been ascribed to the genus *Eunice sensu lato*, because we refer the systematic issue on the distinction between the genera *Eunice* and *Leodice* to the discussion.

Key to identification of Mediterranean *Eunice sensu lato* species:

1. Antennae smooth, short and terminating at the same height. Branchiae starting far from the prostomium from 60°–70° chaetigers; peristomial cirri short not reaching the middle of the anterior peristomium ring . . . . . . . . . . . . . . *Eunice schizobranchia*
   Branchiae starting within the first chaetigers . . . . . . . . . . . . . . . . . . . . . . . . . 2
2. Acicula yellow . . . . . . . . . . . . . . . . . . . . . . . . . . . . . . . . . . . . . . . . . . . 3
   Acicula black . . . . . . . . . . . . . . . . . . . . . . . . . . . . . . . . . . . . . . . . . . . 6
3. Acicular hooks bidentate . . . . . . . . . . . . . . . . . . . . . . . . . . . . . . . . . . . . . 4
   Acicular hooks tridentate . . . . . . . . . . . . . . . . . . . . . . . . . . . . . . . . . . . . 5
4. Antennae indistinctly annulated or smooth; branchiae pectinate with up to 10–12 filaments, starting at the 3° chaetiger and lacking in the posterior part of the body . . . . . . . . . . . . . . . . . . . . . . . . . . . . . . . . . . . . . . . . . . *Eunice pennata*
   Antennae clearly annulated long, with long cylindrical articles; branchiae pectinate with up to 10–16 filaments; first branchiae at 3°–4° chaetiger and to near posterior end . . . . . . . . . . . . . . . . . . . . . . . . . . . . . . . . . . . . . . . *Eunice harassii*
5. Antennae deeply annulated, moniliform with short segments. Branchiae pectinate with maximally 12 filaments, starting from chaetiger 4°–7° and present in the posterior segments . . . . . . . . . . . . . . . . . . . . . . . . . . . . . . *Eunice antennata*
   Antennae indistinctly articulated. Branchiae pectinate with maximally 12–14 filaments. starting from chaetiger 3 and lacking in the posterior segments . . . . . . . . *Eunice vittata*
6. Antennae smooth or indistinctly articulated . . . . . . . . . . . . . . . . . . . . . . . . 7
   Antennae regularly articulated or moniliformis . . . . . . . . . . . . . . . . . . . . . . 8
7. Antennae smooth and short, similar in length reaching the 2°–4° chaetiger, branchiae starting at 6°–10°, often 8°–9°, chaetiger, pectinate with numerous up to 15–40 filaments longer than the notopodial cirri, very large species . . . . . . . . . . . . . . . . . . . . . . . . . . . . . . . . . . . . . . . . . . . . . . . . . . . . . . *Eunice roussaei*
   Antennae indistinctly articulated and long, first branchiae on chaetiger 6°–10° often 7°, pectinate with 3–12 filaments, about as long as the notopodial cirri . . . . . . . . . . . . . . . . . . . . . . . . . . . . . . . . . . . . . . . . . . . . . . . . . . . . *Eunice norvegica*
8. Antennae articulated with cylindrical articulations; branchiae starting from 3°–4° chaetiger, with 2–3 rarely 5–6 filaments . . . . . . . . . . . . . . . . . . *Eunice oerstedi*
   Antennae clearly moniliform . . . . . . . . . . . . . . . . . . . . . . . . . . . . . . . . . . . 9
9. First branchiae on chaetiger 7°–10° often 9°, branchiae pectinate with 4–8 filaments longer than the notopodial cirri; antennae articulated with moniliform increasingly distally articulations . . . . . . . . . . . . . . . . . . . . . . . . . . . . . . *Eunice floridana*
   First branchiae starting before 7° chaetiger . . . . . . . . . . . . . . . . . . . . . . . . 10
10. Branchiae poor developed, palmate with few 1–2 maximally 3 filaments, starting from 3° chatiger . . . . . . . . . . . . . . . . . . . . . . . . . . . . . . . . . . . . *Eunice dubitata*
    Branchiae well developed . . . . . . . . . . . . . . . . . . . . . . . . . . . . . . . . . . . 11
11. Branchiae starting at 2°–3° chaetiger, pectinate with up to 8–10 filaments; body colour pattern uniform bright orange without spots nor whitish ring on anterior segment; subacicular hooks always in single arrangement . . . . . . . . . . . . *Eunice laurillardi*
    Branchiae starting from 3° chaetiger, pectinate with several up to 10–14, often 7, filaments,

body colour pattern red brown with a white collar at the 4° chaetiger; subacicular hooks in double arrangement in medial and posterior segments . . . . . . . . . . . . . . . *Eunice torquata*

## 4. Discussion

Polychaete assemblages associated with the different mesophotic bioconstructions investigated off the Southeastern Italian coast, Apulia, proved to be very rich and diverse in species number and composition, with 70 species recorded overall, most of which are included both in the family Serpulidae and in the order Eunicida. In fact, polychaetes were dominant in terms of number of species compared with the other taxonomic groups associated with the bioconstructions, such as Bryozoans, with 50 species recorded throughout all sites [43], and Porifera with 59 and 65 species collected respectively from the coral reef of MON and from the *Neopycnodonte* bioconstructions of OTR and SML [22,24]. Facing the similar levels of α-diversity, ranging from 40 to 46 species, the polychaete assemblages of the different mesophotic bioconstructions showed a high turnover rate along the geographical North–South gradient as showed by β-diversity and by the similarity Sörensen index, with the highest affinity between southern sites: e.g., *Lepidastenia elegans* and *Harmothoe antiplopes* from MON were substituted by *Harmothoe pagenstecheri* at OTR and SML; *Eunice dubitata* and *E. schizobranchia* characterized MON and OTR instead of *E. pennata* and *E. floridana* which characterized SML; *Syllis alternata* and *S. variegata* from MON were replaced by *Syllis ferrani* and *Haplosyllis spongicola* at OTR; *Filogranula calyculata* and *Filogranula annulata* from OTR and SML took the place of *Filogranula gracilis* from MON. In addition, *Hydroides pseudouncinata*, exclusive to northern sites, was replaced by various species exclusive to southern sites i.e., *Serpula lobiancoi, Plagosteus tridentatus, Josephella marenzelleri, Protula tubularia*. On the basis of these results, it should be hypothesized that the observed increase of β-diversity is driven primarily by the typical hydrological features of the Apulian Adriatic coast [44–48]. This is supported by the circulation pattern of the area, where the surface current gyre flows southeastward, being responsible for moving larvae and propagules in such a direction and, as a consequence, for the different degree of connectivity between the sites (pre-settlement factors). However, this is not the only factor driving benthic community dynamics and population connectivity, because also competitive and facilitating factors are well known to be very influential (see Giangrande et al. [49]). Among the biological determinants, we highlight the relevant role of the bioconstructors, which are primarily responsible for the different bioconstructions, being mainly scleractinians at MON and deep oysters (*Neopycnodonte cochlear*) at OTR and SML [22,24]. In agreement with the above explanation, other studies come to similar conclusions, concerning the bryozoan assemblages from the same area [43] and other taxa from different habitats, e.g., molluscs [50], sponges [51] and brackish waters communities [52,53].

A further result of our study concerns the role of polychaetes in affecting the mesophotic bioconstruction structure. In fact, within the polychaete assemblages studied, we found a relevant functional guild diversity, among which the roles of secondary constructors, binders, dwellers, destroyers-borers may be detected. Serpulids are the main builder worms, which were subordinate in terms of carbonate production to scleractinians and deep oysters, but notwithstanding they contributed to increase the concretion structure, fixing their calcareous tubes in epibiosis on skeletons and valves of the primary constructors. So, they consistently enhanced surface heterogeneity, particularly by the species exhibiting gregarious habit, e.g., *Serpula massiliensis* and *Filograna implexa*, and created interstices and crevices which are suitable for collecting sediment particles close to hard surfaces of the substratum. In this way, these polychaetes acted as facilitators for the colonization of invertebrates with diverse substratum affinities. Some serpulids, e.g., *Serpula* spp., *Hydroides pseudouncinata, Sprirobranchus* spp., with their particular large-sized tubes coated other calcareous structures forming bridges and so they played the role of binders. Other serpulids, with small-sized tubes, e.g., most spirorbids *Protolaeospira (Protolaeospira) striata, Pileolaria militaris, Vinearia koehleri*, were observed on the bare surfaces, thus showing their pioneer role in the colonization pattern.

The growth of serpulids was noted to be essentially linked to the interaction with the other organisms forming the bioconstructions. In fact, they compete for space mainly with other sessile taxa such as bryozoans, which could extensively overgrow the calcareous tubes of the worms. Notwithstanding, the result of such interactions does not always lead to the death of serpulids, which continue to grow below the encrusting bryozoan colonies, like the observation that the tube openings remain free and not covered by the colonies supported. Our study also exhibits that the twisted calcareous tubes of large serpulids, particularly *Serpula cavernicola* and *S. massiliensis*, appeared to be lacking in erosion scars, in contrast to the evidence of large crevices on numerous corallites reported for coralligenous concretions, due to the excavating action of borer organisms such as clionid sponges [54]. These diverse patterns can be explained by differences in substratum microtexture, as the microcrystalline structure exhibited by serpulid tubes [55], as well as in terms of specificity of the boring sponge action [56]. Our observations advanced crucial implications on the role of bioerosion in balancing the growing and destroying phases of the mesophotic bioconstructions that deserve further investigations. A special mention is deserved, within the vagile fauna, to the eunicids, e.g., *Eunice dubitata, E. torquata, E. floridana*, the numerous and large specimens of which were recorded in association with corals and oysters, being directly living among the scleractinian corallites and the valve of *Neopycnodonte*. Unfortunately, this association is not yet clarified and causes some concerns on the nature of opportunistic "nestler" or true "bioeroder" for such species.

In short, in the southern bioconstructions, which are dominated by *N. cochlear*, the contribution of serpulids was more relevant, accounting for 36 species in SML, as compared with the 16 species collected at the northern site of MON, where the bioconstruction is dominated by scleractinians. An opposite situation was exhibited by the vagile fauna particularly with the Eunicida: even only considering the species richness, in fact, the species of this order are quite absent in the southern bioconstructions, with only 3 species at SML, against the 11 species of the northern site of MON, where *E. dubitata* was the dominant species.

## 5. Taxonomic Considerations

Our study revealed some relevant novelty for the polychaetological fauna of Mediterranean Sea concerning the genus *Eunice*. The analysis, based on comparisons of the sampled material with preserved scientific collections and with the relative scientific literature, made it possible to reach an updated baseline for the identification of the species, which have been recorded from the Mediterranean. However, confusion still exists on the synonymy and distribution of some species of the genus *Eunice sensu lato*. The genus is still deserving a clear definition and for this reason we referred all the species in the dichotomous key to *Eunice sensu lato*. In their recent phylogenetic revision Zanol et al. [36] stated the genera *Leodice* and *Nicidion* as resurrected to name monophyletic groups and including species previously ascribed to *Eunice* and *Marphysa*. The authors considered the articulation of prostomial appendages, other prostomial features and the regionalization of the body as characters supporting the monophyly of the family and genus level clades. According to Zanol et al. [36], some Mediterranean species such as *E. antennata, E. harassi, E. torquata, E. laurillardi, E. dubitata, E. floridana*, and *E. vittata* should be ascribed to the genus *Leodice*. By contrast, *Eunice norvegica* and *E. roussaei*, phylogenetically distant from the members of the genus *Leodice*, should be included in the genus *Eunice sensu strictu*. However, some other species, such as *E. oerstedi, E. schizobranchia* and *E. pennata*, need further analyses to clarify their systematic position. About other concerns, below we synthesize the issues on the species mostly subjected to the taxonomic debate. *Eunice roussaei* Quadrefages, 1866 is a very large species originally described on specimens reported from the Antilles Islands. It has been reviewed by Fauvel [57] and more recently by Fauchald [30]. This species has been separated from the similar *E. aphroditois* (Pallas, 1788) on the basis of the paired subacicular hooks disposition in most segments, the shape of pectinate chaetes, the branchiae morphology and the longer antennae reaching the fourth setiger; *E. aphroditois*

has been considered synonym of *E. roussaei* in the checklist of the Italian fauna [41]. Zanol and Bettoso [33] as well proved that the specimens collected in the Adriatic Sea should be referred to *E. roussaei* and not to *E. aphroditois*. In the present study we analysed the specimens of the Giangrande's personal collection coming from Southern Adriatic Sea off the Apulian coast and agreed with the aforementioned Authors, thus we considered *E. roussaei* a valid species of the Mediterranean and Italian fauna and included it in the key to identification.

Some taxonomic confusion also concerns the species *Eunice purpurea* Grube, 1866, whose original description was based on specimens from the lagoon of Lesina, Southern Adriatic Sea. Afterwards, Fauvel [27] considered this species as a juvenile form of *E. roussaei*. Recently, Salazar et al. [35] noted the specimens from Adriatic to be similar to *E. purpurea*, redescribed by Fauchald [30] as a valid species. *E. purpurea* is also reported as synonym of *E. roussaei* in the checklist of the Italian fauna [41] and we agreed with this latter position regarding the synonymy of the two species and with the statement of Zanol and Bettoso [33] concerning the taxonomic debate on the identity of *E. roussaei*.

Regarding other novelty of our study, it should be mentioned the record of *E. dubitata* that was the first for the Mediterranean and the Italian coast and proved this species to be characteristic of the mesophotic bioconstructions, especially those which are primarily built by the non-symbiotic scleractinians, *Phyllangia americana mouchezii* and *Polycyathus muellerae* and by the deep-sea oyster *Neopycnodonte cochlear*.

This study also expands the list of Italian polychaete fauna with the new record of *E. dubitata*, so increasing to 11 the Mediterranean species of the genus *Eunice*.

Lastly, the checklist is also amended as far as the distribution of *E. norvegica*, a species living in association with deep-water white corals, as the recent records from Southern Adriatic and Ionian [16,17,58,59] and Tyrrhenian Sea [60] evidenced. By contrast, the record cited in the checklist from the coralligenous of the Marine Protected Area of Porto Cesareo [61] has to be exactly referred to *E. torquata*.

**Supplementary Materials:** The following are available online at https://www.mdpi.com/article/10.3390/d13060239/s1, Figure S1: Photo of the mandibles of *Lumbrineris coccinea*.

**Author Contributions:** M.F.G., Conceptualization, Data curation, Investigation, Taxononomical and Formal analysis, Writing—original draft, Writing—review & editing, Supervision, C.P., Data curation, Field methodology, Writing—review & editing. M.M., Data curation, Laboratory methodology, Writing—review & editing. C.N.M., Data curation, Investigation, Laboratory metodology, Writing—review & editing. A.G., Conceptualization, Data curation, Investigation, Taxonomical and Formal analysis, Writing—original draft, Writing—review & editing, Supervision. All authors have read and agreed to the published version of the manuscript.

**Funding:** This research received no external funding.

**Institutional Review Board Statement:** Not applicable. Ethical review and approval were waived for this study, due to nor humans or protected animals were involved.

**Informed Consent Statement:** Not applicable.

**Data Availability Statement:** All data generated or analysed during this study are included in this published article.

**Conflicts of Interest:** The authors declare no conflict of interest.

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
