# Peer review of "Polychaete Diversity Related to Different Mesophotic Bioconstructions along the Southeastern Italian Coast"

_diversity, doi:10.3390/d13060239_

Round 1

Reviewer 1 Report

Dear authors,

The study "Polychaete diversity related to mesophotic bioconstruction types along the environmental gradient off the eastern Apulia coast (Italy)" is interesting, despite quite local. Nevertheless,  the scientific work is well organized and written, thus deserving to be published in the journal "Diversity" after addressing the following major comment.

Authors must demonstrate that Lumbrineridae species reported in the present study are in agreement with the recent work on Italian lumbrinerids: https://doi.org/10.1080/11250003.2016.1154615. Lumbrineris coccinea (as well as L. latreilli) is often used misidentified over the European coasts as demonstrated by several recent publications. Please, provide accurate identification and photos of mandibles.

Please, pay attention to the use of italics in Table 1 (e.g., Serpula vermicularis Linnaeus, 1767).

Author Response

We revised the manuscript according to the suggestions posted by the Reviewers 1 and 2 and to the Editor’s comments and now we gladly present a new version of the manuscript with all the changes tracked.

In particular, we amended the text according to the suggestions for increasing search engine discoverability: we have optimized the abstract, placing essential findings and keywords in the first two sentences; we have made the abstract shorter; we have also shortened the title (11 words) also including three keywords; we also included the keywords throughout the text.

Reply to the Reviewer 1

We revised the specimens of the family Lumbrineridae paying particular attention to the mandibles, prostomium shape and setation, according to the paper by D’Alessandro et al. (2016) on Italian lumbrinerids and included the reference in the bibliography. So, we confirm our identification of Lumbrineris coccinea, whose specimens were characterized by the maxillary apparatus bearing five pairs of maxillae, MI as long as MII, MIII clearly bidentate and by composite multidentate hooded hooks with short blade.

Reviewer 2 Report

Review

Paper title: Polychaete diversity related to mesophotic bioconstruction types along the environmental gradient off the eastern Apulia coast (Italy).

Biodiversity of the mesophotic zone in the Mediterranean Sea is poorly studied. The authors sampled benthic assemblages at 3 sites along the Adriatic and Ionian coast of Apulia to describe the species composition of the local polychaete fauna. The authors found some differences in the diversity and attributed these to the environmental conditions of the sites. The authors provided an updated key to identification of the species belonging to the genus Eunice. The first record Eunice dubitata adds this species to the list of the Mediterranean and the Italian fauna.

All these reasons explain the relevance of the paper by Maria Flavia Gravina and co-authors submitted to "Diversity".

General scores.

The data presented by the authors are original and significant. All conclusions are justified and supported by the results. The study is correctly designed and technically sounds. In general, the statistical analyses are performed with good technical standards. We authors conducted careful work which will attract the attention of a wide range of specialists focused on the benthic ecology and diversity as well as taxonomists.

Recommendation.

L 85-86. The authors should provide additional details for sampling stations: sampling period (date), environmental conditions (temperature, salinity, etc.).

Specific comments.

L 16-18. Change “In the present study the diversity patterns, in terms of species composition and of functional role, of polychaete assemblages of three bioconstructions reported for the Apulian coast were analysed; the update” to “In the present study, the species composition and functional role of polychaete assemblages were analysed; the updated”

L 20. Change “the total of 70 species. Facing the similar  levels” to “a total of 70 species. Facing similar  levels”

L 31. Change “bioindicators for” to “bioindicators of”

L 32. Change “of sedimentary and rocky bottoms.” to “on sedimentary and rocky bottoms”

L 35. Change “object” to “an object”

L 39. Change “sea-mountains” to “sea mountains”

L 42. Change “, the so-called” to “so-called”

L 45. Change “Ligurian sea” to “the Ligurian Sea”

L 46. Change “sea” to “Sea”

L 52. Change " components' to " component"; “This taxon” to “This group”

L 60. Change “of functional” to “functional”

L 63. Change “genus” to “the genus”

L 71. Change “the northernmost and is” to “northernmost and”

L 79. Change “nautical mile” to “nautical miles”

L 83. Change “in all three” to “at all three”

L 88. Change “in laboratory” to “in the laboratory”, " alcohol solution" to "an alcohol solution"

L 99. Change “disentangling” to “disentangle”

L 105. Change “and of” to “and”

L 107. Change “being S” to “where S is”

L 109. Change “being” to “is”; " of sample" to " of a sample".

L 120. Change “specie” to “species”

L 125. Change “, resulted” to “were”

L 128. Change “typically inhabitant” to “typical inhabitants”

L 137. Change “species which” to “which”

L 143. Change “little species” to “small species”

L 152. Change “as result” to “as a result”

L 157. Change “collected abundant” to “abundant”

L 159. Change “found abundant in” to “abundant at”

L 161. Change “P. siciliensis” to “and P. siciliensis”

L 164. Change “total length” to “a total length of”

L 165. Change “segments respectively” to “segments, respectively”

L 174. Change “the minimum” to “a minimum”

L 175. Change “the maximum” to “a maximum”

L 179. Change “particular” to “particularly”

L 182. Change “found associated to” to “associated with”

L 184. Change “found associated to” to “associated with”

L 185. Change “resulted” to “was”

L 189. Change “Ionian” to “the Ionian”

L 196. Change “update key” to “updated key”

L 198. Change “of peristomial” to “peristomial”

L 205. Change “associated  to the  mesophotic  concretions  investigated” to “associated  with the  mesophotic  concretions ”

L 206. Change “Apulian coast” to “the Apulian coast”

L 209. Change “associated to” to “associated with”

L 213. Change “number of species” to “species”

L 223. Change “base” to “basis”

L 225. Change “to the typical hydrological features of Apulian” to “by the typical hydrological features of the Apulian”

L 228. Change “consequence” to “a consequence”

L 229. Change “one factor” to “factor”

L 230. Change “the biological” to “other”

L 231. Please, clarify what are these " post-settlement factors"?

L 233. Change “responsible of” to “responsible for”

L 241. Change “Serpulids resulted” to “Serpulids are”

L 242. Change “in term” to “in terms”

L 248. Change “like facilitators for” to “as facilitators for the”

L 250. Change “large size” to “large-sized”

L 252. Change “little size” to “small- sized”

  1. 254 " The growth of serpulids…" should be the beginning of a new paragraph.

L 257. Change “; these latter” to "which”

L 258. Change “do not always produce” to “does not always lead to”

L 259. Change “to growth” to “to grow”; "as the observation" to " like the observation"

L 260. Change “tubes openings” to “tube openings”

L 262. Change “contrast with” to “contrast to”

L 264. Change “borers organisms” to “borer organisms”

L 271. Change “associated” to “in association”

L 276. Change “role of serpulids resulted” to “contribution of serpulids was”

L 284. Change “comparison” to “comparisons”

L 286. Change “made possible” to “made it possible”

L 289. Change “species included” to “species”

L 311. Change “Apulian” to “the Apulian”

L 316. Change “juvenile form” to “a juvenile form”

L 327. Change “increases” to “expands”.

Author Response

Dear Editor,

we revised the manuscript according to the suggestions posted by the Reviewers 1 and 2 and to the Editor’s comments and now we gladly present a new version of the manuscript with all the changes tracked.

In particular, we amended the text according to the suggestions for increasing search engine discoverability: we have optimized the abstract, placing essential findings and keywords in the first two sentences; we have made the abstract shorter; we have also shortened the title (11 words) also including three keywords; we also included the keywords throughout the text.

Reply to Reviewer 2.

We agree with Reviewer 2 and provided the required additional details for sampling site in the paragraph 2.2. ”Sampling methods and taxonomic analysis”. Moreover, we amended the text according to all the specific comments suggested. We made all revisions easily visible being marked by “Track Changes” function.

Round 2

Reviewer 1 Report

Dear authors,

Unfortunately, you do not provide any picture that could validate your identification as L. coccinea. Assuming that the team is right, I'd like to kindly alert you that the identification of MIII in lumbrinerids is particularly challenging. Is quite normal to get mistaken between a MIII bidentate and MIII unidentate followed by a knob. These re-identifications have been happening  and discussed in numerous recent publications across the European coast, because several lumbrinerid species are misidentified. I recall you that several lumbrinerid species were discovered in the last 10-20 years in Europe highlighting the huge effort to update the diversity of the family Lumbrineridae, but also the past and recent difficulties to identify these organisms due to the lack of training on the very complex mandible structure.

Please, for the sake of clarity I insist and recommend them once again to recheck some animals, compare with the literature and, provide one good picture of the MIII of such specimens. It would clearly benefit the ms.

All the best

Author Response

We agreed with Reviewer 1 that Lumbrineris coccinea is often used misidentified over the European coasts, so we paid particular attention to identification and included in the manuscript the diagnostic characters which led to identification of our specimens (see lines 191-196). On the other hand, we considered inappropriate to add the photo of mandibles in the paper, but we included it in our reply.

In addition, we corrected the use of italics in Table 1 (Serpula vermicularis, Linnaeus, 1767).  
